# NOT ONLY VISION: EVOLVE VISUAL SPEECH RECOGNITION VIA PERIPHERAL INFORMATION

## ABSTRACT

Visual Speech Recognition (VSR) aims to infer what was said by analyzing the speaker's facial dynamics. However, is reliance solely on visual information sufficient in challenging real-world scenarios? In human visual perception, peripheral vision refers to non-central areas of the visual field, crucial for providing overall awareness and detailed perception of central objects. Similarly, human lip-readers do not rely exclusively on lip movements but integrate contextual cues and prior knowledge to achieve more accurate transcribing. For the first time in machine lip-reading, we frame these non-lip-movement factors into a new concept of semantic-level peripheral information, Specifically, we select three representative types varying in relevance to the spoken content: (1) *Contextual peripheral information*, such as the general topic or some basic knowledge of the speech, can significantly narrow the range of potential recognition hypotheses. (2) *Experiential peripheral information* emerges from the recognition process itself. The very act of recognizing speech in a specific language provides implicit knowledge of grammar, word collocations, and related linguistic aspects, thereby guiding the recognition effectively. (3) *Perturbative peripheral information* introduces disturbance factors into the recognition process, analogous to noise injection in visual tasks. Semantic-level peripheral information is indirectly linked to transcripts; thus fusing it into VSR necessitates strong contextual understanding and inference capabilities. Here, we propose a multimodal learning framework built on a large language model (LLM), leveraging its powerful contextual modeling capabilities to take advantage of peripheral information. Our method's efficacy is demonstrated on two popular datasets. On the widely-used LRS3 dataset, we achieved a Word Error Rate (WER) of 24.5% with readily available peripheral information, leading to an impressive 14.3% relative improvement over the model without such information. To the best of our knowledge, our work sets a new state-of-the-art when utilizing similar hours of lip-reading videos. We further reported the evaluation on the more challenging AVSpeech dataset. Results across both datasets and various experimental settings demonstrate the promising potential of the proposed semantic-level peripheral information for VSR.

## 1 INTRODUCTION

Visual Speech Recognition (VSR) analyzes the visual dynamics of lip movements during speech to infer spoken content. It has gained wide attention due to its important applications, including assisting speech interpretation in noisy environments(Sumby & Pollack, 1954), and offering a communication method for patients with speech disabilities (McGurk & MacDonald, 1976). However, VSR still faces significant challenges due to the scarcity of labeled video data and the complexity of real-world scenarios.

Existing methods primarily focus on enhancing the utilization of visual information from the lip region. Approaches involving constructing and utilizing larger lip-reading datasets(Afouras et al., 2018a; Serdyuk et al., 2022; Chang et al., 2024), or employing self-supervised pre-training techniques(Shi et al., 2022; Haliassos et al., 2022; Zhu et al., 2023; Zhang et al., 2024), have achieved significant improvements in VSR. However, is optimizing visual information processing the only path towards robust VSR in complex real-world scenarios? Lip-reading is also a challenging task even for human lip-reading experts. Observations of human lip-reading suggest that the process involves more than just visual dynamics.

Drawing inspiration from peripheral vision in the field of human visual perception, we introduce the concept of **Peripheral Information** to reduce the gap between machine and human lip-reading capabilities. The human visual system comprises both central and peripheral vision. Peripheral vision, the visual field beyond our current point of gaze (Vater et al., 2022), plays a critical role in providing an overall awareness and detailed perception of central objects by capturing the broader context(Larson & Loschky, 2009). For instance, as illustrated in the Figure 1, when focusing on the central vision area, it's challenging to identify the object. However, with information from peripheral vision, we are sure that it is a window on an old building.

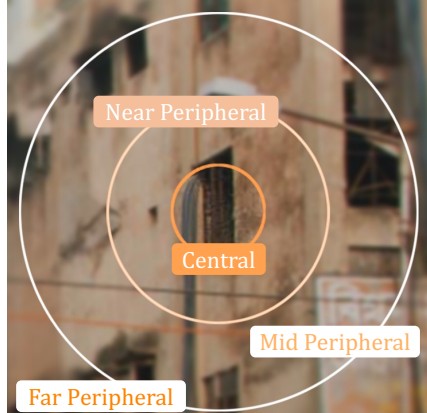

Figure 1: Example of peripheral vision.

Similarly, human lip-readers intuitively utilize what we term as peripheral information to facilitate recognition. Peripheral information includes various aspects, such as a rough estimation of the potential content, awareness of the scenarios, and their own expertise. For instance, having a general idea about the topic being discussed can help lip-readers predict likely words and phrases, reducing ambiguity. Moreover, experienced lip-readers possess a keen awareness of potential errors and actively engage in self-correction. They leverage available cues about the speaker or the context to rectify misinterpretations critically. This cognitive awareness helps them identify and refine potential errors more effectively.

We identify three representative and easily accessible types of peripheral information in semantic level based on the relevance to the speech content: contextual, experiential, and perturbative. And we systematically explore their contribution to VSR. Given that these information sources often have indirect correlations with the content to be recognized, effectively utilizing them poses a significant challenge for traditional VSR methods. Large Language Models (LLMs) have demonstrated remarkable capabilities in contextual understanding and reasoning across diverse domains. Given these powerful abilities, we propose a framework built on a LLM, harnessing its powerful contextual modeling and reasoning abilities to incorporate these peripheral information sources into the recognition process.

We evaluated the effectiveness of our proposed approach on two widely recognized datasets: the LRS3 TED dataset(Afouras et al., 2018b) and the AVSpeech dataset(Ephrat et al., 2018) derived from YouTube content. Our experiments on the LRS3 dataset showed impressive improvement, with the best Word Error Rate (WER) of 24.5% when incorporating readily accessible peripheral information. This represents a substantial 14.3% relative improvement compared to the baseline model without such information. Notably, even in scenarios where only limited contextual peripheral information, e.g. the scenario information of the speech, was utilized, we still observed a significant performance boost, achieving 9.1% relative WER reduce. To our knowledge, these results establish a new benchmark for VSR performance when considering similar amounts of training lip-reading videos. Furthermore, our evaluation extends to the more complex AVSpeech dataset, where we observed consistent improvements across configurations. These comprehensive findings across both datasets underscore the potential of semantic-level peripheral information in enhancing visual speech recognition performance.

## 2 RELATED WORK

### 2.1 VISUAL SPEECH RECOGNITION

VSR has evolved significantly driven by advancements in computer vision and machine learning techniques. Traditional VSR approaches relied heavily on hand-crafted features and statistical models (Petajan et al., 1988; Goldschen et al., 1997; Luettin & Thacker, 1997). The advent of deep learning marked a paradigm shift in VSR technology in recent years. Convolutional Neural Networks (CNNs)(LeCun et al., 1995) and Recurrent Neural Networks (RNNs)(Hochreiter, 1997;

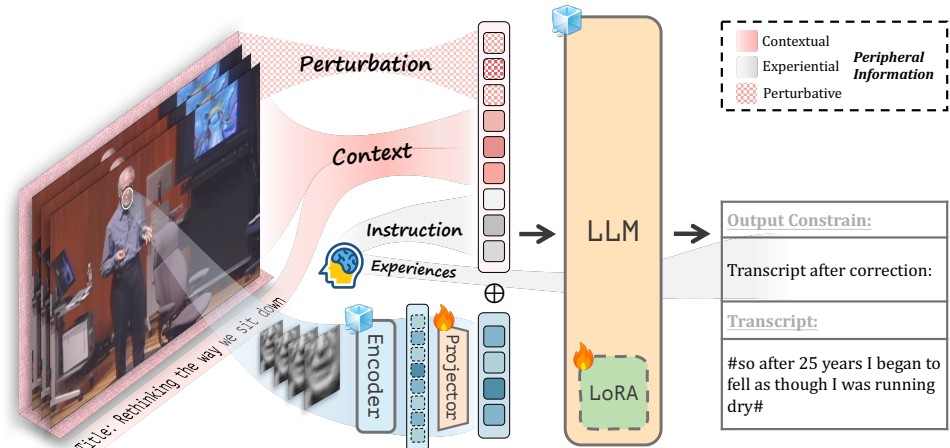

Figure 2: Proposed framework for integrating peripheral information into visual speech recognition. Visual speech representations from encoder are pooled and mapped to the embedding space of LLM. The encoder and LLM are kept frozen while projecotr and LoRA are learnable.

Chung et al., 2014) have been widely adopted to capture both spatial and temporal aspects of lip movements (Assael et al., 2016; Chung & Zisserman, 2017; Petridis et al., 2017; 2018; Yang et al., 2019). The transformer (Vaswani, 2017) has further advanced the field by effectively capturing long-range dependencies in lip movement sequences (Afouras et al., 2018a; Ma et al., 2021b; 2023). Self-supervised methods(Shi et al., 2022; Haliassos et al., 2022) have learned robust speech representations. These deep learning models have shown remarkable improvements in recognition accuracy compared to traditional methods.

VSR still remains challenging due to the inherent ambiguity in lip movements despite these advancements. This ambiguity frequently leads to recognition errors, especially for words with similar pronunciations. Relying solely on lip dynamics is exceedingly difficult to overcome such ambiguity. Consequently, the integration of contextual information becomes not just beneficial, but crucial for achieving more accurate and reliable recognition. Previous VSR research has made impressive advancements in processing visual dynamics of lip region. However, while the importance of contextual information is recognized, its integration into VSR systems remains limited. In this study, we introduce the concept of peripheral information, which encompasses a broader scope than traditional contextual information, and systematically investigates the contribution of it.

## 2.2 CONTEXT-AWARE SPEECH RECOGNITION

In the field of auditory speech recognition (ASR), researchers have investigated various methods to incorporate contextual information into ASR systems. Several works(Chan et al., 2016; He et al., 2017; Jain et al., 2020; Huber et al., 2021; Sathyendra et al., 2022) introduce extra context biasing module into end to end ASR system to process phrase-level contextual information. More recent approaches(Radford et al., 2023; Lai et al., 2023; Lakomkin et al., 2024; Nozawa et al., 2024) directly incorporate broader contextual information, such as previous utterances or keywords, into ASR systems without specialized biasing modules. However, in the field of VSR, related work is neglected to the best of our knowledge.

The concept of peripheral information in VSR that we propose extends existing approaches. While previous methods primarily focus on integrating content-related contextual information, our study goes beyond these conventional cues. We explore a more comprehensive set of elements from various aspects, ranging from contextual ones to experiential information from human-lip-reading process and disturbance factors, aiming to provide a broader perspective for improving VSR.

# 3 METHOD

Our study seeks to enhance VSR by introducing information beyond visual dynamics, which we term as peripheral information in this paper. It encompasses a wider range of information surrounding VSR more than content-related contextual cues. We take three representative types of easily accessible semantic peripheral information and explore their contribution to VSR, We seamlessly incorporate them into the process of VSR by leveraging the contextual modeling abilities of the LLM. Figure 2 illustrates the overall framework of our method, which achieves our core idea through the following two key aspects:

- **VSR-adapted LLM** commits to enable a LLM with the ability to process visual speech representations and generate transcripts in a certain format, which is achieved by modality adaptation and LLM specialization.
- **Peripheral Information Fusion** aims to achieve jointly inference for the target transcripts using visual speech features together with the semantic peripheral information via the VSR-adapted LLM. We explore a simple manner of structured prompt and guided generation to evaluate the role of semantic peripheral information for VSR.

## 3.1 VSR-ADAPTED LLM

### 3.1.1 MODALITY ADAPTATION

We transform the input visual sequence to enable processing by the LLM, which is pretrained on textual data. Visual sequences differ from LLMs' representation space in temporal, spatial, and semantic dimensions. Visual sequences have a significantly higher temporal resolution compared to the token-based representations used in LLMs. Furthermore, visual data is inherently different from that of textual data in dimension and semantics. Therefore, for the combination of semantic-level peripheral information and visual dynamics in a unified framework, appropriate modality adaptation is necessary.

To be specific, given an input video sequence of lip region $X_v = \{x_i\}_{i=1}^T$ with $T$ frames at 25 fps. We utilize the widely used AV-HuBERT(Shi et al., 2022), a self-supervised pre-trained model for speech representations, as visual encoder to obtain robust visual speech representations. The encoder AV-HuBERT ($\mathcal{E}$) extracts initial visual speech features as:

$$F_v = \{f_i\}_{i=1}^T = \mathcal{E}(X_v) \in \mathbb{R}^{T \times D_\mathcal{E}},$$

where $D_E = 1024$ is the feature dimension of the encoder. The initial visual speech representations are at 25 fps, which is considerably higher than the token rate of corresponding transcript. This high temporal resolution significantly increases computational demands. To balance temporal information preservation and computational efficiency, we apply a $2\times$ average pooling operation for temporal down-sampling to obtain the zipped visual features $Z_v$.

$$Z_v = \{z_i\}_{i=1}^{\lfloor \frac{T}{2} \rfloor} = \text{AvgPool}(F_v, k = 2, s = 2).$$

To bridge both semantic and dimensional gap between visual and textual representations, we employ a simple linear projector, mapping $Z_v$ to the LLM's embedding space:

$$E_v = Z_v W \in \mathbb{R}^{\lfloor \frac{T}{2} \rfloor \times D_L},$$

where $D_L = 4096$ is the dimensionality of the LLM's embeddings and $W \in \mathbb{R}^{D_E \times D_L}$ is the weight matrix of the linear projector. By employing this simple modality adaptation, we enable LLM to process visual speech inputs.

### 3.1.2 LLM SPECIALIZATION

To adapt LLM's linguistic modeling capabilities to the specialized task of VSR, we employ Low-Rank Adaptation (LoRA) (Hu et al., 2022) to efficiently fine-tune the LLM. By introducing learnable rank decomposition matrices together with a cross-modal alignment objective to the original

backbone network weights, specific adaptation to the target task is allowed with few additional parameters. We apply LoRA to the query and value weight matrices in the self-attention layers of the LLM. For each weight matrix $W_{Attn.}$, LoRA introduces a low-rank update which leads to $W'_{Attn.} = W_{Attn.} + \Delta W_{Attn.} = W_{Attn.} + BA$, where $B$ and $A$ are small learnable matrices. The original LLM parameters are all kept frozen and only LoRA matrices are learnable.

To guide the LLM to process visual speech features and generate textual transcript. We introduce a conditioned cross-entropy loss for the probability distribution output by the LLM. Let $I$ denote the instruction *Transcribe the speech*, and $Y = \{y_i\}_{i=1}^n$ the transcript of $n$ words. The instrcution and visual features are concatenated as the input of the LLM: $H = [I; E_v]$. The loss is calculated as:

$$\mathcal{L} = -\sum_{i=2}^n \log P(y_i | I, E_v, \{y_j\}_{j=1}^{i-1}).$$

where $P(y_i | I, E_v, \{y_j\}_{j=1}^{i-1})$ is the conditional probability distribution from the LLM's output layer.

## 3.2 PERIPHERAL INFORMATION FUSION

This paper introduces a new framework for Visual Speech Recognition (VSR) that goes beyond traditional lip-movement analysis. Inspired by peripheral vision in human visual systems, we propose the concept of peripheral information in VSR, which includes a wider range of information We identify three types of peripheral information: contextual, experiential, and perturbative. While contextual information has been used in speech recognition, our approach expands its application in VSR. Importantly, we introduce experiential and perturbative information as new categories, which differ from existing concepts in the field. These information types have distinct sources and roles in VSR, requiring specific integration strategies. This framework aims to provide a new perspective on visual speech recognition. In the following subsections, we will introduce the three types of peripheral information and present our methods for integrating them into our VSR-Adapted LLM.

### 3.2.1 CONTEXTUAL PERIPHERAL INFORMATION

Contextual Peripheral Information (CPI) denotes a wide range of background factors that can provide relevant context, helping to infer the speech content. This information consists of elements such as the scenario, title or an overview of the target talk or video, or a brief introduction of the speaker. Take LRS3 as an example, table 1 presents the available CPI types for it, along with their availability ratios, and examples. The availability ratio represents the percentage of samples in the dataset containing each type of information. It's important to note that all CPI types provide information about the video as a whole, rather than a specific utterance within it.

Table 1: Overview of Contextual Peripheral Information (CPI) of LRS3.

| Type | Availability (Proportion of samples) | Example |
|---|---|---|
| Scenario | 100% | A speech from TED talk |
| Speaker name | 30% | Niels Diffrient |
| Speaker description | 30% | Designer |
| Talk title | 73% | Rethinking the way we sit down |
| Talk description | 73% | Design legend Niels Diffrient talks about his life in industrial design... |

Emulating the human ability to grasp the overall context of a speech and so to infer the content more accurately, we structured available CPI in form of natural language into the input of LLM. This approach unifies various types of CPI and harnesses the semantic understanding capabilities of the LLM to effectively utilize this contextual information. For instance, *The following is {scenario} named {title}. The description of the talk is {description}* naturally combines scenario and descriptive information of the speech. If certain CPI is missing for a certain sample, we use *The following is {scenario}* as default. To enable the model to consider and utilize this information, we modified the above loss function. Let $C$ denote the structured contextual peripheral information. The input to the LLM is the concatenation of CPI, the instruction $I$, and the visual embedding $E_v$, denoted

as $H = [C; I; E_v]$. We modify the learning objective to predict accurate transcripts conditioned on CPI, which means minimizing the following loss function:

$$\mathcal{L}_c = -\sum_{i=2}^{n} \log P(y_i|C, I, E_v, \{y_j\}_{j=1}^{i-1}).$$

### 3.2.2 EXPERIENTIAL PERIPHERAL INFORMATION

Experiential Peripheral Information (EPI) complements our approach by incorporating higher-level information related to the speech recognition process. While CPI provides valuable content-related context, EPI focuses on aspects of the recognition task itself. Emulating the human ability to self-reflect during speech recognition, we introduce a dual-component mechanism that encourages the model to engage in self-correction. The first component is an active prompt that guides the model's generation process. We implement this by modifying the instruction $I$ to $\hat{I}$: *Transcribe the speech and then correct possible errors*. This prompts the model to engage in self-correction when generating transcript. The second component introduces a constraint on the model's output. We prefix the objective transcript $Y$ with *Transcript after correction*, resulting in $\hat{Y} = \{\hat{y}_i\}_{i=1}^{m}$ of length $m$. By calculating loss on all content of the prefixed transcript $\hat{Y}$, we explicitly constrain the model to generate self-corrected transcripts. The input for LLM is reformulated as $H = [C; \hat{I}; E_v]$. To formalize this approach, we refine the loss function to include CPI and EPI as follows:

$$\mathcal{L}_{c,e} = -\sum_{i=2}^{m} \log P(\hat{y}_i|C, \hat{I}, E_v, \{\hat{y}_j\}_{j=1}^{i-1}),$$

By minimizing $\mathcal{L}_{c,e}$, we train the model to further incorporate EPI, encouraging itself to explicitly acknowledge its correction step and ultimately produce more accurate transcripts.

### 3.2.3 PERTURBATIVE PERIPHERAL INFORMATION

Perturbative Peripheral Information (PPI) represents disturbance factors in the speech recognition process. Analogous to noise injection in visual tasks, we explore PPI as a form of semantic perturbation to enhance of the model's robustness. In contrast to contextual and experiential peripheral information, PPI introduces semantic disturbances unrelated to speech content, allowing us to investigate their impact on recognition performance.

To operationalize this concept, we incorporate token-level perturbations into the input sequence. To maintain the integrity of the original input $H$ for LLM when introducing disturbance, we prefix the input with random tokens. This approach allows us to inject noise without directly altering or disrupting the meaning of the subsequent sentences. Specifically, we prefix the input with a random number of random tokens. Let $R = \{r\}_{i=1}^{k}$ denote a sequence of $k$ random tokens, where $k$ itself is a random variable determining the perturbation length. The modified input sequence becomes $H = [R; C; \hat{I}; E_v]$. To condition the model's output alongside CPI, EPI and PPI, we adjust the loss function to:

$$\mathcal{L}c, e, p = -\sum_{i=2}^{m} \log P(\hat{y}_i|R, C, \hat{I}, E_v, \{\hat{y}_j\}_{j=1}^{i-1}).$$

## 4 EXPERIMENTS

### 4.1 SETUP

Our experiments were primarily conducted on LRS3(Afouras et al., 2018b), a widely-used benchmark for visual speech recognition. To further validate our approach in more complex real-world scenarios, we also present results on the AVSpeech(Ephrat et al., 2018) dataset. LRS3 comprises thousands of spoken sentences from TED and TEDx videos providing both visual and audio components, along with corresponding transcripts. AVSpeech is a large-scale audio-visual dataset extracted

Table 2: **Results on LRS3.** We compare the WER of our method with prior works. When utilizing similar hours of unlabelled (Unlab.) and labelled (Lab.) hours of data, our method with peripheral information (PI) outperforms prior methods.

| Method | Unlabelled Hours | Labelled Hours | WER (%) |
|---|---|---|---|
| *Fully Supervised Models* | | | |
| Zhang et al. (2019) | - | 863 | 60.1 |
| Ma et al. (2021b) | - | 595 | 43.3 |
| Prajwal et al. (2022) | - | 698 | 40.6 |
| Ma et al. (2022) | - | 1,459 | 31.5 |
| Ma et al. (2023) | - | 3,448 | 19.1 |
| *Self-supervised Pre-training & Supervised Fine-tuning* | | | |
| Ma et al. (2021a) | 1,759 | 433 | 38.8 |
| Shi et al. (2022) | 1,759 | 433 | 28.6 |
| Zhu et al. (2023) | 1,759[1] | 433 | 28.4 |
| Haliassos et al. (2022) | 1,759 | 433 | 27.8 |
| Yeo et al. (2024b) | 1,759 | 433 | 27.6 |
| Yeo et al. (2024a) | 1,759 | 433 | 26.7(25.4[2]) |
| A: B - LoRA[3] | 1,759 | 433 | 27.4 |
| B: Ours | 1,759 | 433 | 26.6 |
| C: B + EPI | 1,759 | 433 | 26.2 |
| D: C + PPI | 1,759 | 433 | 26.0 |
| E: D + Limited CPI[4] | 1,759 | 433 | 25.6 |
| F: C + Rich CPI[5] | 1,759 | 433 | **24.5** |
| *Trained using non-publicly available datasets* | | | |
| Afouras et al. (2018a) | - | 1,519 | 58.9 |
| Shillingford et al. (2018) | - | 3,886 | 55.1 |
| Serdyuk et al. (2022) | - | 90,000 | 17.0 |
| Liu et al. (2023) | 3,652 | 3,068 | 16.9 |
| Chang et al. (2024) | - | 100,000 | 12.8 |

[1] Uses additional 3846h unpaired audio, 452h audio-text and 600M unpaired text data.
[2] Includes additional fine-tuning of the encoder.
[3] Is the model with projector layer trainable only, and without any peripheral information.
[4] Only provides scenario information.
[5] Does not use PPI for it decrease performance when using rich CPI, see Section 4.3 PPI.

from YouTube, representing a broader range of real-world speaking scenarios. We utilized the English portion of it to test the robustness of our method. As AVSpeech originally lacked transcriptions, we employ Whisper(Radford et al., 2023) for automatic annotation. To maintain consistency in evaluation, we trimmed the test set of AVSpeech to match the duration of the LRS3 test set.

Contextual peripheral information is collected from readily available sources. For LRS3, we used a pre-collected dataset from Kaggle [1]. It contains information about all talks including descriptions, speakers and titles and is originally collected for visually exploring TED Talks statistics. However, as this dataset only covered 30% of the samples in LRS3, we collected additional data from available YouTube links. The data statistics are presented in Table 1. For AVSpeech, we collected video title and description for approximately 97% of the samples directly from YouTube links. To minimize computational burden, we limited descriptions within three sentences or 100 words (whichever came first) and replaced website links with a generic `url` to avoid extremely long ones.

In the implementation of our framework, a lip-centered 96x96 pixel region of interest (ROI) of each video frame was taken as the raw input, and the output of the final layer of the AV-HuBERT was utilized as the initial visual speech representations. We adopt the pre-trained LLaMA-2 7B(Touvron et al., 2023) as the backbone of VSR-Adapted LLM. For LLM specialization, we used LoRA with

---

[1]https://www.kaggle.com/datasets/thegupta/ted-talk/data

a rank of 256 (see Appendix A.2 for comparison), alpha of 512, and dropout of 0.1. The model was trained for 16 epochs using an AdamW optimizer under default settings and a reciprocal learning rate scheduler. During inference on the test set, we employed beam search with a width of 8. For evaluation, we use word error rate (WER) as the evaluation metric. All training and evaluation were conducted on 8 NVIDIA A100 40GB GPUs.

## 4.2 MAIN RESULTS

Table 2 presents comparisons of our method with previous works on the LRS3 test set. Compared to AV-HuBERT(Shi et al., 2022), which is used as our encoder, our VSR-adapted LLM reduced the WER from 28.6% to 26.6% (A), demonstrating the effectiveness of modality adaptation and LLM specialization.

Then we fuse three types of peripheral information, each contributing to improved performance. Experiential peripheral information (EPI), emulating human self-correction through a dual-component mechanism, yielded a 0.4% (from B to C) absolute reduction in WER.This improvement suggests that human experiential cognition in lip reading, when utilized as peripheral information, can also benefit machine-based approach.

Perturbative peripheral information (PPI) introduces disturbance factors as a form of peripheral information into the VSR task. Building upon the EPI, the additional incorporation of PPI yielded a slight improvement (from C to D). Although modest, this improvement demonstrates that semantic-level PPI can enhance the robustness of VSR.

We progressively incorporating contextual peripheral information (CPI). Even limited CPI, providing only scene information, reduced the WER to 25.6% (E). The richer variant, including titles and descriptions of TED talks, achieves our best result of 24.5% WER (F), surpassing previous methods using similar hours of data. These results strongly demonstrate the advantage of leveraging peripheral information in machine lip reading.

Table 3: WER on the AVSpeech under different PI configurations. In this table, S. stands for Scenario CPI, T. for video title, and D. for video description.

| PI Config | w\o | EPI & PPI | S. & T. | S. & T. & EPI | S. & T. & PPI | S. & T. & EPI & PPI | S. & T. & D. |
|---|---|---|---|---|---|---|---|
| WER (%) | 49.4 | 49.0 | 47.8 | 47.7 | 46.9 | **46.8** | 47.7 |

Results on AVSpeech are shown in Table 3. These experiments extend our evaluation to a more challenging and realistic environment, further validating the effectiveness of our proposed Peripheral Information integration approach.

## 4.3 CONTRIBUTION OF PERIPHERAL INFORMATION

To gain a more comprehensive understanding of our approach, we further evaluated the contribution of the three types of semantic-level peripheral information to VSR.

Table 4: WER on the LRS3 test set using different CPI types. In this table, S.N. stands for speaker name, S.D. for speaker description, T.T. for talk title, and T.D. for talk description. 100 and unltd refer to 100-word-limited and unlimited talk description, respectively. ASP stands for availability sample proportion. Row A includes results without EPI and PPI, B includes results with both.

| CPI type | w\o | Scenario | S. N. | S. D. | T. T. | T. T. | T. D. (100) | T. D. (100) | T. D. (unltd) |
|---|---|---|---|---|---|---|---|---|---|
| ASP (%) | - | 100 | 30 | 30 | 30 | 73 | 30 | 73 | 73 |
| A | 26.6 | 26.7 | 25.9 | 26.5 | 26.4 | 26.3 | 25.2 | **24.7** | 25.6 |
| B | 26.0 | 25.6 | 25.8 | 25.9 | 25.8 | 25.4 | 25.5 | 24.9 | 26.3 |

**Contextual Peripheral Information (CPI)** encompasses a diverse range of auxiliary data. We evaluated the contribution of each available CPI type in our framework, with results shown in Table 4 row A. Simple CPI types, such as speaker names, descriptions and talk titles, provide concise identifying information, offering modest improvements in VSR accuracy. Among these, speaker

names show a more significant performance boost. Speakers in TED are often well-known figures, and their names help pinpoint their areas of expertise, thereby narrowing down potential vocabulary associated with each speaker's field. This focused CPI allows the model to achieve lower WER.

More complex CPI type, the talk descriptions, demonstrate stronger performance gains. The 100-word limited description achieved the best performance with a WER of 24.7%. Unlimited-length descriptions, while still beneficial, showed a slight decrease in performance. Longer descriptions often contained promotional content, less relevant to speech recognition. This finding highlights the importance of concise, targeted contextual peripheral information in enhancing VSR performance.

Adding scene information slightly decreased performance here. This concise, high-level information seems challenging for the model to effectively utilize, possibly introducing noise rather than valuable context. The performance decrease observed when using talk descriptions will be further analyzed in the sections discussing EPI and PPI.

In summary, these findings underscore the crucial role of CPI in VSR. Different types of CPI contribute to performance improvements in varying degrees, with more focused and relevant information generally yielding better results.

**Experiential Peripheral Information (EPI)** is implemented through a dual-component mechanism to stimulate the model's self-correction capability. The effects of EPI under varying CPI conditions are summarized in the corresponding columns of Table 5. It consistently improves performance across all levels of CPI, with WER reductions ranging from 0.2% to 0.6% demonstrating its effectiveness and robustness in enhancing VSR.

The most substantial gains are observed when only limited CPI with only the scenario provided, where EPI significantly reduces WER from 26.7% to 26.1%, surpassing the setting without CPI (26.2%). This result demonstrates that EPI it helps the model overcome the difficulties in interpreting abstract, high-level context through proposed self-correction mechanism.

As more detailed CPI is integrated, the relative impact of EPI diminishes, but it continues to contribute to performance improvements. The best overall performance at a WER of 24.5% is achieved when EPI is combined with rich CPI, including scenario, title and description.

These findings highlight the unique advantages of our proposed EPI approach. Unlike traditional methods that rely solely on external context, EPI leverages the model's own predictions to enhance performance. This self-correcting mechanism proves particularly effective in scenarios with limited or inconsistent CPI, demonstrating EPI's robustness and adaptability.

Table 5: WER under different CPI configurations with EPI and PPI. The numbers under PPI indicate the random token length range. The values in the table represent WER (%).

| CPI | | | EPI | | PPI (w\EPI) | | | |
|---|---|---|---|---|---|---|---|---|
| Scenario | Title | Description | w\o | w\ | 1-5 | 3-9 | 6-18 | 10-50 |
| ○ | ○ | ○ | 26.6 | 26.2 | 26.0 | 26.0 | 26.1 | 26.0 |
| ✓ | ○ | ○ | 26.7 | 26.1 | 25.6 | 25.7 | 25.7 | 27.7 |
| ✓ | ✓ | ○ | 26.3 | 25.8 | 25.7 | 25.7 | 25.4 | 25.6 |
| ✓ | ✓ | ✓ | 24.7 | **24.5** | 25.3 | 25.3 | 25.2 | 25.7 |

**Perturbative Peripheral Information (PPI)** introduces semantic disturbance into the VSR process. We investigated the impact of perturbative non-lip-movement factors on the model's robustness, implementing by prefixing random tokens to the input. We investigated its impact across various mean lengths, building upon previous results incorporating CPI and EPI. The effects of PPI under different CPI configurations, are summarized in the PPI-related columns of Table 5.

The setting with only of EPI combined with PPI yields marginal WER improvement. However, when limited CPI providing only scene information is introduced, PPI demonstrates more substantial gains. Notably, when random tokens with a length range of 1-5 are added, a significant WER improvement of 0.5% is achieved. And as comparison between row A and B in Table 1 demonstrate,

integrating with EPI and PPI makes scene information contributes positively to performance. This suggests PPI's effectiveness, as well as its combination with EPI.

Conversely, when scenario, title, and description are provided, the introduction of PPI leads to performance degradation. Results from Table 4 also show the same phenomenon. This suggests that when effective contextual information is both sufficient and diverse, additional perturbation becomes superfluous. To gain more insight into this phenomenon, we conducted experiments with scenario and talk description CPI, results are shown in Table 6.

Table 6: Effect of Talk Description (T.D.) length and PPI on WER. (100) and (unltd) refer to 100-word-limited and unlimited talk description. PPI (6-18) means prefixing random tokens with lengths ranging from 6 to 18.

| Config | T.D. (unltd), w\o PPI | T.D. (100), w\o PPI | T.D. (100), w\ PPI (6-18) |
|---|---|---|---|
| WER (%) | 24.9 | 24.7 | 24.9 |

The performance achieved unlimited-length description was comparable to that obtained using a description limited to 100words supplemented with PPI. This observation implies that excessively lengthy contextual information may be unnecessary. PPI can compensate for insufficient contextual information, while becoming redundant when peripheral information is adequate. Furthermore, for extensive contextual information, methods for selective filtering should be explored.

To conclude, PPI effectively increases model robustness by introducing random tokens, demonstrating its value in enhancing adaptability to diverse input conditions, especially when dealing with limited or abstract contextual information.

Table 7: Comparison of different EPI variants. Constrain-only EPI includes instruction $I$ in model input $H$, while the left use $\hat{I}$. Gray words does not contribute to loss calculation.

| EPI Variant | Output Constrain (Loss Scope) | WER (%) |
|---|---|---|
| Complete EPI | Transcript after correction: {transcript} | **26.2** |
| No-Loss Constrain EPI | Transcript after correction: {transcript} | 27.0 |
| Implicit Constrain EPI | The transcript should be: {transcript} | 26.3 |
| Prompt-only EPI | #transcript# | 27.1 |
| Constrain-only EPI | Transcript after correction: {transcript} | 27.2 |

**Components in EPI** To analyze the role of the two components in EPI, we conducted comparative experiments with five configurations with results shown in Table 7. The results reveal that both the instructive prompts and generative constraints are crucial elements of the EPI framework. This approach leverages the strengths of large language models, which often perform better when guided to think step-by-step through complex tasks, thus improving the overall performance of the model.

## 5 CONCLUSION

We introduce Peripheral Information (PI), a novel concept in visual speech recognition that leverages non-lip-movement information, and propose an LLM-based multimodal learning framework that integrates three representative types into the recognition process. Our experiments on the widely used lip-reading dataset LRS3 and the more complex real-world dataset AVSpeech demonstrate the effectiveness of PI. Notably, we achieved state-of-the-art performance on LRS3 using comparible hours of lip-reading video. We conducted detailed experiments on different PIs to explore their contributions to VSR, elucidating the contribution of and the fusion strategies for different types of PI. We hope our study inspires future research in and beyond visual speech recognition.

ETHICAL STATEMENT

This study utilizes publicly available datasets, including LRS3 and AVSpeech, for research purposes. All data used in this paper are publicly available and are used under the following licenses: the TED terms of use, the Creative Commons BY-NC-ND 4.0 license and Creative Commons Attribution 4.0 International License. Data used in this study encompass a wide array of speakers, but they may not fully capture global diversity. Our review shows a good mix of genders, ethnicities, and age groups. However, the makeup of the data might not exactly match the world's population spread. This means our model's effectiveness could vary when applied to different groups of people. Additionally, despite numerous positive applications, our method can potentially be misused such as surveillance, compromising public privacy and trust. Appropriate regulatory measures need to be established to address these concerns.

REPRODUCIBILITY STATEMENT

To ensure reproducibility, we provide as many implementation details as possible in the paper. All data and models used are publicly available, with their sources clearly documented in Section 4.1.

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

# A APPENDIX

## A.1 DATA

Table 8: Statistical summary of CPI used in LRS3. (100) refers to limmit talk description in 100 words or three sentences (which comes first).

| Type of CPI | Length | | |
|---|---|---|---|
| | 25% Quantile | Mean | 75% Quantile |
| Talk Name | 4 | 6.2 | 8 |
| Talk Description | 58 | 127.4 | 182 |
| Talk Description (100) | 34 | 45.5 | 56 |
| Speaker Name | 2 | 2.2 | 2 |
| Speaker Description | 1 | 2.1 | 3 |

Table 9: Statistical summary of CPI used in AVSpeech.

| Type of CPI | Length | | |
|---|---|---|---|
| | 25% Quantile | Mean | 75% Quantile |
| Title | 6 | 8.4 | 11 |
| Description | 33 | 115.2 | 140 |

Table 8 and Table 9 summarize the length statistics of various types of Contextual Peripheral Information (CPI) used in the LRS3 and AVSpeech dataset. They provides the 25th percentile, mean, and 75th percentile lengths for each CPI type, offering an overview of the contextual information available for our VSR experiments.

## A.2 LoRA Rank

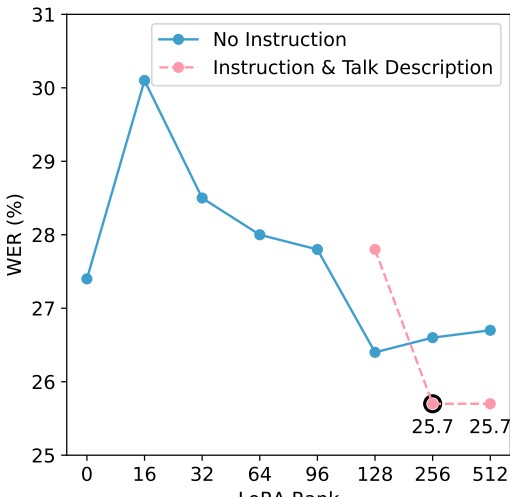

Figure 3: WER under different LoRA rank settings

While previous works applying large language models to speech recognition have utilized LoRA with predetermined ranks, we conducted a systematic exploration of different LoRA ranks for optimizing the trade-off between model capacity and computational efficiency.

Figure A.2 presents the WER for different LoRA ranks, with and without contextual PI. Without peripheral information, WER initially increases with LoRA rank, then decreases, reaching a minimum around rank 128. This suggests an optimal point for effective VSR task learning without overfitting.

When fusing TED talk abstracts as contextual PI, WER increases at rank 128 compared to the no-PI scenario. However, at ranks 256 and 512, WER decreases and stabilizes, outperforming the no-PI condition. This indicates that rank 256 provides sufficient capacity to handle both the VSR task and additional information effectively.

Based on these findings, we chose a LoRA rank of 256 for subsequent experiments, offering the best performance-efficiency trade-off in our multimodal VSR framework.

## A.3 Downsampling strategy

Table 10: WER results for different downsampling rates and methods.

| Downsampling Rate | Method | WER (%) |
|---|---|---|
| - | - | 27.6 |
| 2× | Concatenation | 28.5 |
| 2× | Average Pooling | **27.4** |
| 3× | Average Pooling | 29.9 |

We employed downsampling strategy to address the computational challenges and temporal discrepancy with text arising from the high frame rate of visual features. Experiments are conducted without LoRA or peripheral information, revealing that average pooling with a 2× average pooling provides the optimal balance between performance and computational efficiency. As shown in Table 10, it slightly outperforms the baseline (WER 27.4% vs. 27.6%) while significantly reducing computational costs. Higher downsampling rates or alternative methods like simple concatenation led to performance degradation, indicating the importance of preserving fine-grained temporal information.

## A.4 PROMPT EXAMPLES

Table 11: Prompt examples for integrating CPI and EPI

| LRS3: w\o CPI, w\o EPI |
| --- |
| Transcribe the speech. |
| LRS3: w\ scenario CPI, w\o EPI |
| A speech from TED talk. Transcribe the speech. |
| LRS3: w\ scenario, talk name CPI, w\o EPI |
| A speech from TED talk named Rethinking the way we sit down. Transcribe the speech. |
| LRS3: w\ scenario, talk name CPI, w\o EPI |
| A speech from TED talk named Rethinking the way we sit down. The description of the talk is "Design legend Niels Diffrient talks about his life in industrial design (and the reason he became a designer instead of a jet pilot). He details his quest to completely rethink the office chair starting from one fundamental data set: the human body." Transcribe the speech. |
| LRS3: w\o CPI, w\ EPI |
| Transcribe the speech and then correct possible errors. |
| LRS3: w\ scenario CPI, w\ EPI |
| A speech from TED talk. Transcribe the speech and then correct possible errors. |
| AVSpeech: w\ scenario CPI, w\ EPI |
| A speech from a YouTube video titled Great Lesson Ideas: The Iditarod & Math. Transcribe the speech and then correct possible errors. |

Table 11 shows example prompts used in our example.

