# OpenReview forum: "Not Only Vision: Evolve Visual Speech Recognition via Peripheral Information"
_ICLR.cc/2025/Conference — ICLR 2025 Conference Withdrawn Submission_

### Official Review · Reviewer_CCh9 · 2024-11-01

**Soundness:** 3
**Presentation:** 4
**Contribution:** 3
**Rating:** 6
**Confidence:** 5

**Summary:**

This paper proposes the idea of using peripheral information in addition to visual information in an attempt to improve the performance of visual speech recognition models. Three different types of peripheral information are proposed: contextual, experiential and perturbative. The proposed framework is based on LLMs and and evaluated on the LRS3, achieving SOTA results, and AVspeech datasets.

**Strengths:**

The paper is well written and easy to follow.

The idea of using peripheral information for VSR is novel. To the best of my knowledge, it's the first time it is applied to VSR.

State-of-the-art results are presented on LRS3.

Several ablation studies are presented.

**Weaknesses:**

The ablation study showing the effect of each type of information (bottom of Table 2) could have been better designed. It would be better
if components were removed from F, i.e. F - EPI, F - PPI, F - CPI instead of the current approach where each component is added on top of the previous one. The proposed ablation study would clearly show the benefit of each type of information.

The AV-Hubert is used as a visual encoder, however it's not the state-of-the-art model anymore.  Why not using other SOTA visual encoders like the ones from AV-Data2vec or Raven or AutoAVSR? Alternatively, why not using multiple visual encoders? This could be another useful ablation study which would show that the proposed approach is beneficial independently of the visual encoder.

It is not clear why the perturbations added to enhance the model's robustness are considered as peripheral information. It's a type
of augmentation. Can the authors please elaborate on this?

It's not clear why the AVspeech test set is trimmed to the same size as the LRS3 dataset. Why not using the full test set?

Line 373, it would be good to add some additional information in the appendix on how the mouth ROI is cropped? e.g, are facial landmarks first detected? If yes, how?

There are some typos, some examples are the following:
Caption of Fig. 2, projecotr -> projector
line 310, funcition -> function
line 356, Is the model with projector layer trainable only, I think something is missing in this sentence

**Questions:**

Please see the weakness section.

---

### Official Review · Reviewer_fTux · 2024-11-06

**Soundness:** 2
**Presentation:** 2
**Contribution:** 2
**Rating:** 3
**Confidence:** 3

**Summary:**

This work introduces a novel framework for visual speech recognition (VSR) that leverages semantic-level peripheral information, inspired by human peripheral vision, to enhance lip-reading capabilities. The proposed method demonstrates significant improvements in word error rate (WER) on the LRS3 and AVSpeech datasets, highlighting its potential to bridge the gap between human and machine lip-reading performance.

**Strengths:**

1）The paper introduces the novel concept of incorporating semantic-level peripheral information into Visual Speech Recognition (VSR), broadening the scope of traditional lip-reading methods by leveraging contextual, experiential, and perturbative information.

2）The proposed framework effectively integrates peripheral information with a large language model (LLM), demonstrating advanced contextual modeling and reasoning capabilities to enhance VSR performance.

**Weaknesses:**

1.	Although the paper introduces the use of peripheral information to aid in lip-reading, it does not provide a comprehensive exploration of the valuable types of peripheral cues that might impact VSR performance. Theoretically, the introduction of additional information should enhance expressiveness, but there is a lack of key exploration into peripheral contextual information, such as identifying which peripheral information is useful, which is not, how to extract useful information, and filter out the useless information. Additionally, how peripheral information actually aids lip-reading, and the differences before and after its use, should be addressed.


2.	The reliance on large language models (LLMs) for processing and integrating peripheral information may limit the scalability and applicability of the method in resource-constrained environments. The computational cost and efficiency of using LLMs have not been thoroughly discussed.

3.	The datasets used, although popular, may not fully capture the diversity of real-world scenarios, such as different accents and languages.

4.	The paper does not provide access to the code used for experiments, which hinders the ability of other researchers to validate the findings and reproduce the results.

5.	In the 2023 paper "Jointly Learning Visual and Auditory Speech Representations from Raw Data," Table 2 shows results on LRS3 with Unlab hours 1,759 + Lab hours 433 that match the data used in your Table 2. That work achieved results of 24.4 without using a language model and 23.1 with it. You used additional data and a large language model, yet your result is 24.5. I find this result to be inadequate.

**Questions:**

1.How does the paper address the challenge of identifying and leveraging valuable peripheral cues for VSR, and what methodologies could be employed to better distinguish between useful and non-useful peripheral information?

2.What considerations have been made regarding the computational cost and efficiency of using large language models (LLMs) for integrating peripheral information, especially in resource-constrained environments?

3.Is there a possibility for the code used in this study to be made publicly available to enhance reproducibility and validation of the findings by the research community?

---

### Official Review · Reviewer_8cAd · 2024-11-06

**Soundness:** 3
**Presentation:** 2
**Contribution:** 2
**Rating:** 5
**Confidence:** 4

**Summary:**

This work introduces peripheral information to enhance lip reading by identifying three types—contextual, experiential, and perturbative—and integrating them using large language models (LLMs). The proposed method demonstrates improvements in word error rate (WER) on the LRS3 and AVSpeech datasets.

**Strengths:**

The paper introduces experiential peripheral information (EPI), marking the first time peripheral information is incorporated into the perspective of lip-reading, whereas previous work primarily used cropped lip ROI regions as model input.

By incorporating peripheral information, the method achieves relative improvements in Word Error Rate (WER) on two lip-reading datasets, setting a new state-of-the-art benchmark for similar durations of lip-reading videos.

**Weaknesses:**

The innovation of integrating peripheral information into lip-reading is insufficient. Previous research has conducted similar studies, comparing the performance of lip-reading using raw video frames, cropped faces, cropped lips and surrounding areas, and cropped lips.

Figure 1: Example of peripheral vision is not representative and has no relation to lip-reading.

The paper "Large Language Models Are Strong Audio-Visual Speech Recognition Learners" is not cited, yet it is highly similar to your model and achieves better performance. With a large language model and visual lip-reading on LRS3, it reports a WER of 24.0, using AV-HuBERT with Trainable Parameters (M) = 48. Thus, I believe the performance in this paper is inadequate, especially since additional data was used for training. I would like to know if the trainable parameters in this paper exceed 48.

**Questions:**

Which peripheral information is useful and which is not? Concrete examples should be provided, and it should be clearly explained how peripheral information affects lip-reading.

Can you train the comparison methods with peripheral information and compare their performance with the method proposed in this paper?

What is the number of trainable parameters (M) in this paper?

**Details Of Ethics Concerns:**

Facial video data was used.

---

### Official Review · Reviewer_5trX · 2024-11-08

**Soundness:** 3
**Presentation:** 1
**Contribution:** 2
**Rating:** 3
**Confidence:** 5

**Summary:**

The paper proposes to use external (periphery) non-lip related cues to improve the performance of lip reading. For instance, one can provide information on the general topic being spoken about and the name of the speaker. Using these additional cues, an improvement of 2.2 WER points is observed. Details analysis is conducted to study the impact of various pieces of such periphery information.

**Strengths:**

Clear improvement in scores. Motivation is strong — it is well-known that lip reading can improve a lot with more context and this work explores this aspect on a large scale for the first time.

**Weaknesses:**

The paper tries to build a story around “peripheral vision” and introduces a lot of unintuitive terminologies and abbreviations. The presentation is unnecessarily made difficult to understand, despite the idea of the paper being very straightforward— improving lip reading by providing additional context such as topic, title, and the name of the speaker.

The paper also does not compare with a couple of important baselines. Lip reading models have used external LLMs to improve the WER. [1] combines the logits of an LLM (trained on the dataset) and the lip reading model’s logits while decoding. [2] re-ranks the beam search predictions using a pre-trained LLM.

Another experiment that is missing is how important the visual information is for correcting this transcript? How well can you do if you just had the outputs?
Can you take only the transcription from a pre-trained lip reading model (instead of retraining it along with peripheral context), provide the additional peripheral information to an LLM and Lora-fine-tune it to predict the corrected transcription?

The paper is also missing more analysis to understand the cases where such peripheral information helped to improve the transcription and where it did not.

[1] https://arxiv.org/pdf/2102.06657
[2] https://openaccess.thecvf.com/content/CVPR2022/papers/Prajwal_Sub-Word_Level_Lip_Reading_With_Visual_Attention_CVPR_2022_paper.pdf

**Questions:**

Providing more details on the missing experiments/baselines would encourage me to raise my rating.

---

### Note · Authors · 2024-11-15

I have read and agree with the venue's withdrawal policy on behalf of myself and my co-authors.